# CLERF: Contrastive LEaRning for Full-Range Head Pose Estimation

**Ting-Ruen Wei**[1*]**, Huei-Chung Hu**[2*]**, Haowei Liu**[1]**, Xuyang Wu**[1]**, Yi Fang**[1] **and Hsin-Tai Wu**[2*]

[1] Santa Clara University, Santa Clara, CA, USA
[2] DOCOMO Innovations, Sunnyvale, CA, USA
[*] Equal Contribution

## ABSTRACT

We propose a novel framework for representation learning in head pose estimation (HPE) that overcomes the challenges posed by sparse head pose data, which previously made triplet sampling infeasible. Leveraging recent advances in 3D-aware generative adversarial networks (3D GANs), we generate anchor–positive–negative triplets and perform contrastive learning on extensively augmented data, including geometric transformations. This enables the network to learn robust, geometry-aware representations that improve HPE accuracy. We observe that existing HPE models struggle when test images are slightly rotated or flipped, while our method maintains strong performance. Experiments show that our framework matches state-of-the-art models on standard test sets and outperforms them on augmented and full-range poses. Our model handles full-range HPE, accurately predicting head poses across the entire rotation spectrum, including upside-down orientations, and outperforms existing full-yaw range methods.

## 1 INTRODUCTION

Contrastive learning has been widely adopted in computer vision, particularly in unsupervised and self-supervised settings (Chen et al., 2020). However, its use in full-range (FR) head pose estimation (HPE) has received little attention. One practical reason is that contrastive learning relies on meaningful positive pairs, which are difficult to obtain when head poses are sparsely distributed in the three-dimensional rotation space. As a result, most existing HPE methods do not exploit contrastive objectives, especially when considering extreme head orientations.

HPE is a challenging and important problem in computer vision (Hempel et al., 2024; Cao et al., 2020; Zhou & Gregson, 2020; Cobo et al., 2024). The task aims to infer the three-dimensional orientation of a human head from an image and is central to applications such as augmented and virtual reality, driver monitoring, and human–robot interaction. Most existing approaches focus on a limited frontal range, typically restricting yaw angles to $[-90°, 90°]$ (Yang et al., 2019; Huang et al., 2020; Ruiz et al., 2018; Dhingra, 2022; Hsu et al., 2018; Cao et al., 2020; Dai et al., 2020; Zhang et al., 2023). In contrast, FR HPE, which we define as $[-180°, 180°]$ for yaw, pitch, and roll (Wikipedia contributors, 2024), remains underexplored (Zhou & Gregson, 2020; Hempel et al., 2024). Moreover, currently available FR models are still limited in practice and often fail to reliably handle extreme cases such as upside-down head poses.

A fundamental challenge in applying contrastive learning to FR HPE is the sparsity of head pose samples in the rotation space. In a full three-dimensional pose range, it is extremely unlikely to find two real images with nearly identical head orientations. For example, even when allowing a tolerance of $20°$ along each rotational axis, the probability of finding a valid positive sample is approximately $1/18^3 \approx 2 \times 10^{-4}$. This sparsity makes standard triplet or contrastive sampling strategies impractical when relying solely on real-world data. From a geometric perspective, head pose lies on a continuous rotation manifold, where nearby orientations should correspond to nearby representations. However, existing datasets do not provide sufficient coverage to enforce this structure during learning.

In addition to data sparsity, we find that many existing HPE models are sensitive to minor geometric perturbations at test time, such as small image rotations or flips. Even slight deviations from the

training distribution can lead to noticeable performance degradation, suggesting that the learned representations do not adequately capture the underlying geometric structure of head pose. This issue becomes more pronounced in unconstrained scenarios involving dynamic motion, such as sports or acrobatic actions, where extreme poses occur naturally.

To address these challenges, we propose **CLERF**, a contrastive learning framework for FR HPE. CLERF leverages a 3D-aware generative adversarial network (GAN) to synthesize head images with controlled pose parameters. Given a real image, we generate a synthetic head with matching yaw and pitch, and apply geometric transformations to align the synthetic image with the exact head orientation of the real image. This process guarantees the availability of anchor–positive pairs, enabling effective contrastive learning over the full pose range. The use of synthetic data allows dense sampling of rare and extreme orientations, while geometric transformations further expand pose coverage beyond what is typically observed in real datasets. Through contrastive learning and extensive geometric augmentation, CLERF learns pose-aware representations that respect neighborhood structure in the rotation space, leading to improved robustness and accuracy across the full range of head poses.

Our main contributions are summarized as follows:

- We demonstrate how 3D-aware GANs can be used to generate reliable anchor–positive pairs, enabling contrastive learning for full-range head pose estimation.

- We derive and apply general geometric transformations that allow synthetic images to precisely match real head poses and expand coverage across the full three-dimensional rotation space.

- We show that many existing HPE models are sensitive to minor geometric perturbations at test time, including small rotations and flips.

- We achieve performance on par with state-of-the-art methods on standard test sets and outperform them under slight image augmentations and extreme pose variations.

- We present a true FR HPE model capable of accurately predicting arbitrary head poses, including upside-down orientations, and demonstrate superior performance compared to existing full-yaw-range approaches. Code will be released.

## 2 RELATED WORK

**Head pose estimation.** Existing HPE methods can be broadly divided into landmark-based and landmark-free approaches. Traditional landmark-based methods estimate pose from detected facial features, with Dlib (King, 2009) being a representative example. While effective under frontal views, these methods degrade when landmark detection fails, particularly at large yaw angles. As a result, deep learning approaches have largely shifted towards landmark-free models. Early methods, such as HopeNet (Ruiz et al., 2018) and WHENet (Zhou & Gregson, 2020), directly regress Euler angles, but this formulation suffers from discontinuities due to ambiguous angle representations (e.g., $0°$ vs. $360°$).

To address this issue, more recent methods predict continuous rotation representations. 6DRepNet and 6DRepNet360 (Hempel et al., 2024) regress a $3 \times 2$ rotation representation, while TriNet (Cao et al., 2020) predicts a full $3 \times 3$ rotation matrix, with Euler angles computed only for evaluation. Other works focus on improving robustness and generalization: Kuhnke & Ostermann (2023) enforce relative pose consistency to bridge the synthetic-to-real domain gap; Opal (Cobo et al., 2024) aligns reference coordinate systems and introduces a geodesic loss; SemiUHPE (Zhou et al., 2024) leverages unlabeled data via semi-supervised learning with augmentations; and TokenHPE (Zhang et al., 2023) adopts a transformer-based formulation by predicting orientation tokens.

**Head pose datasets.** Existing datasets have limitations for FR HPE. 300W-LP (Zhu et al., 2016) provides frontal-pose images with Euler angles obtained via 3D morphable models (Blanz & Vetter, 2003), while the CMU Panoptic Dataset (Joo et al., 2017) captures multiple individuals from cameras spanning a hemisphere, allowing wider pose coverage. WHENet (Zhou & Gregson, 2020) transformed Panoptic into a larger HPE dataset, but FR extreme poses, including upside-down heads, remain scarce. These limitations motivate our use of synthetic images and geometric transformations to cover the full head pose space.

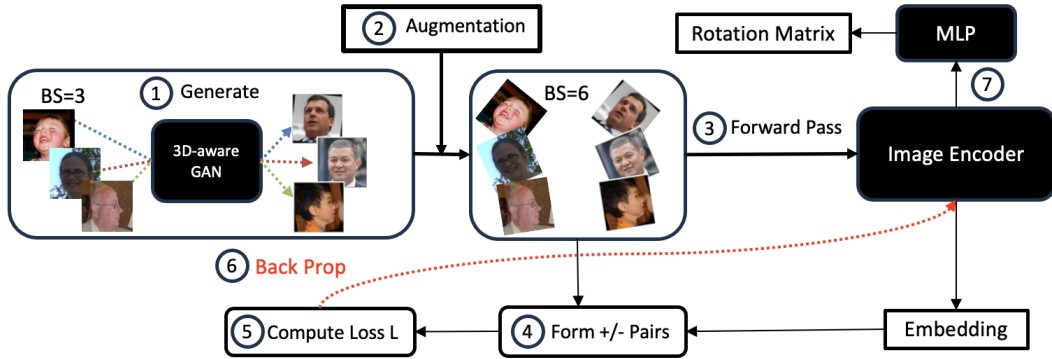

Figure 1: Overview of the proposed contrastive learning framework for full-range head pose representation, illustrated with a batch size of three for clarity. Steps 1–6 constitute one training iteration of the representation model, which is frozen during downstream MLP training in Step 7.

**Contrastive learning.** Lian et al. (2024) employs contrastive learning to align 2D and 3D keypoints in a shared embedding space, and further leverages a graph convolutional network to model geometric relationships for improved HPE. More broadly, related works demonstrate its utility for learning geometry- and pose-aware representations in adjacent domains. In gaze estimation, GazeCLR (Jindal & Manduchi, 2023) and CRGA (Wang et al., 2022) used multi-view contrastive objectives to improve cross-domain generalization. In 3D pose tasks, PeCLR (Spurr et al., 2021) enforced equivariance for hand pose, while P-HLVC (Schneider et al., 2022), Honari et al. (2022), and ICON (Xu et al., 2023) leveraged human pose dynamics and keypoint consistency. In face recognition, PCL (Liu et al., 2023) disentangled pose from appearance. These works show that contrastive learning effectively captures geometric structure, motivating our approach for FR HPE.

## 3 METHODOLOGY

Our proposed methodology involves three key components: anchor-positive generation to empower contrastive learning, geometric transformation to cover the full range of space, and contrastive learning to strengthen the fine-grained head pose predictions. An overview is shown in Figure 1.

### 3.1 ANCHOR-POSITIVE SYNTHETIC IMAGE GENERATION

Given a training sample, we generate a synthetic image with the same yaw and pitch values using a 3D-aware GAN. To match the roll angle of the real image (Figure 1, Step 1), we first solve for a triad of $(yaw, pitch, \phi)$ from the rotation matrix of the real image and apply it to the rotation matrix $R \in SO(3)$ (Hu et al., 2024). The new rotation associated with the roll angle $\phi$ is given by:

$$R_{rotated}(\phi) = R_{extrinsic}(\phi) \times R$$
$$= \begin{bmatrix} cos(\phi) & sin(\phi) & 0 \\ -sin(\phi) & cos(\phi) & 0 \\ 0 & 0 & 1 \end{bmatrix} \times R \tag{1}$$

where $R_{rotated}$ is the rotation matrix of the rotated image.

### 3.2 GEOMETRIC TRANSFORMATION FOR FULL-RANGE HPE

The second key component involves applying geometric transformations to head pose images and their corresponding rotation matrices to enable FR HPE (Figure 1, Step 2). Following the mathematics in Hu et al. (2024), we perform a flip across a line $\ell_\theta$ on the XY-plane. Given a rotation matrix $R \in SO(3)$, the resulting rotation after the flip is:

$$R_{flip}(\theta) = Flip\_\theta \times R \times Flip\_X$$
$$= \begin{bmatrix} cos(2\theta) & sin(2\theta) & 0 \\ sin(2\theta) & -cos(2\theta) & 0 \\ 0 & 0 & 1 \end{bmatrix} \times R \times \begin{bmatrix} -1 & 0 & 0 \\ 0 & 1 & 0 \\ 0 & 0 & 1 \end{bmatrix} \qquad (2)$$

where $\theta$ is the counter-clockwise angle between the positive x-axis and the line $\ell$ across which the image and rotation are flipped. Several commonly used geometric augmentations arise as special cases of Equation 2. In particular, flipping about both image axes can be written as:

$$R_{\mathrm{bf}} = \begin{bmatrix} -1 & 0 & 0 \\ 0 & -1 & 0 \\ 0 & 0 & 1 \end{bmatrix} \times R, \qquad (3)$$

Additional symmetric flips and rotation–flip combinations, derived from Equations 1 and 2, are summarized in Appendix A. The impact of our rotation and flip augmentations on the head pose distribution is visualized and discussed in Appendix B (Figure 5). These geometric augmentations help fill gaps in the original dataset, expanding coverage to a wider range of head poses.

### 3.3 CONTRASTIVE LEARNING FRAMEWORK

Our contrastive learning framework uses geometric augmentations and structured triplet sampling for FR HPE.

**Preserving geodesic distance.** To maintain valid positive pairs under augmentation, the same rotation and flip transformations are applied to both images. Let $H$ be a geometric transformation on images, and $A, B \in SO(3)$ be two rotation matrices. The geodesic distance $d$ is preserved under $H$:

$$d(A, B) = d(H(A), H(B)).$$

A proof is provided in Appendix C. Consequently, applying $H$ to a triplet $(a, p, n)$ produces a new valid triplet $(H(a), H(p), H(n))$ for contrastive learning.

**Triplet sampling procedure.** Based on the above property, we construct anchor-positive-negative triplets as follows:

1. Select an image $I_A$ as the anchor and generate a synthetic image $I_P$ using a 3D-aware GAN with rotation matrix $R' = R_{\mathrm{pitch}} \times R_{\mathrm{yaw}}$.

2. For anchors with labeled rotation matrices $R$, decompose $R$ into the triad $(yaw, pitch, roll)$:

$$R = R_{\mathrm{roll}} \times (R_{\mathrm{pitch}} \times R_{\mathrm{yaw}}),$$

   then rotate the synthetic image $I_P$ by $R_{\mathrm{roll}}$ to produce a positive image with rotation matrix $R_{\mathrm{roll}} \times R' = R$ (see Figure 6 in Appendix D for examples).

3. Apply additional random rotation or flip augmentations $H$ to any existing triplet $(a, p, n)$ to generate new valid triplets $(H(a), H(p), H(n))$.

This procedure enables the sampling of a large number of valid triplets and their augmented variants for contrastive learning.

**Embedding and contrastive loss.** After forming a batch of images, we pass them through the image encoder $E$ to obtain embeddings (Figure 1, Step 3). Triplets are formed using the following procedure:

- Anchor-positive pairs are guaranteed by the synthetic image generation. Additionally, neighboring head orientations in the batch with geodesic similarity $\cos(d(R, R'))$ exceeding a threshold $T_{GD}$ are also included as positives (Figure 1, Step 4).

- Hard and semi-hard negatives are selected based on a margin $v$ computed from the Euclidean distance between embeddings.

Finally, we compute the Circle loss (Sun et al., 2020) (Figure 1, Step 5–6) to maximize similarity between positive pairs and minimize similarity between negatives. The loss reweights contributions to emphasize less-optimized samples, improving representation learning.

Once the representation model is trained, it is frozen and a downstream multi-layer perceptron (MLP) is trained to map embeddings to fine-grained head pose angles, represented as a rotation matrix (Figure 1, Step 7). To ensure the output is a valid rotation matrix, the six-dimensional MLP output is converted to a $3 \times 3$ matrix via the Gram-Schmidt process, following 6DRepNet (Hempel et al., 2024). The trained representation and MLP are then used to evaluate test datasets and compare against baseline models.

## 4 EXPERIMENTS

### 4.1 DATASETS

**Training.** We use 300W-LP (Zhu et al., 2017), which contains 122,450 images drawn from multiple databases, predominantly featuring frontal head poses. To enable contrastive learning in the full 3D rotation space, we employ PanoHead (An et al., 2023), a 3D-aware GAN, to generate synthetic anchor–positives. The generated images are used exclusively for representation model training.

**Testing.** Following prior work, we evaluate our models on AFLW2000 (Zhu et al., 2017) and BIWI (Fanelli et al., 2013), which primarily contain frontal faces with yaw angles in the range $[-90°, 90°]$. To assess FR head pose performance, we additionally construct four augmented variants of these datasets: slightly-augmented (SA) AFLW2000, SA BIWI, fully-augmented (FA) AFLW2000, and FA BIWI. The SA variants are generated by rotating each image clockwise by $10°$ and flipping along a line $85°$ counter-clockwise from the positive $x$-axis (see Figure 3, second row). For the FA variants, images are randomly rotated between $-180°$ and $180°$ and flipped along a line between $0°$ and $90°$ counter-clockwise from the positive $x$-axis (see Figure 3, third row).

### 4.2 EXPERIMENTAL SETUP

We train our FR HPE model using a Swin Transformer-based image encoder and a downstream MLP that maps embeddings to rotation matrices. Training uses a combination of real and synthetic images with geometric and pixel-wise augmentations to improve robustness. We compare against several existing HPE models, including FSA-Net, HopeNet, TokenHPE, 6DRepNet, and WHENet. All models are evaluated using mean absolute error (MAE) on yaw, pitch, and roll, with the average of the three (Mean) reported as the primary metric. Full implementation details, including architecture, hyperparameters, and augmentation procedures, are provided in Appendix E.

## 5 EXPERIMENTAL RESULTS

### 5.1 MAIN EVALUATION

Table 1 reports the performance of all models on six test datasets. On AFLW2000, CLERF achieves performance nearly on par with 6DRepNet, trailing by only $1.4°$ in Mean. On BIWI, it matches TokenHPE, falling short by $0.01°$ in Mean. Since CLERF is trained for FR coverage, it optimizes across the entire pose space, leaving slightly less focus on the frontal range where non-FR models concentrate their capacity.

To better assess frontal-range performance, we evaluate on slightly-augmented datasets (SA AFLW2000 and SA BIWI), where test images are rotated and flipped but remain within the frontal range. CLERF outperforms all baseline models, improving over the runner-up by $0.57°$ and $1.29°$ in Mean for SA AFLW2000 and SA BIWI, respectively, demonstrating robustness and generalization beyond the specific angles seen by non-FR models.

We further evaluate FR capability using fully-augmented datasets (FA AFLW2000 and FA BIWI), where images are rotated and flipped across the entire $[-180°, 180°]$ range. CLERF remains the top

Table 1: Performance comparison with baseline models across six datasets. CLERF achieves on-par performance with baselines on the original AFLW2000 and BIWI test sets, and outperforms all baselines on their slightly augmented (SA) variants. On full-range evaluations with heavily rotated images, CLERF surpasses other full-range models by a large margin. FR indicates whether a model supports full-range prediction, and the best results are highlighted in bold.

| Model | FR | Yaw | Pitch | Roll | Mean | Yaw | Pitch | Roll | Mean |
|---|---|---|---|---|---|---|---|---|---|
| | | \multicolumn{4}{c}{AFLW2000} | | | BIWI | | |
| FSA-Net | × | 4.50 | 6.08 | 4.64 | 5.07 | 4.27 | 4.96 | 2.76 | 4.00 |
| HopeNet | × | 6.47 | 6.56 | 5.44 | 6.16 | 5.17 | 6.98 | 3.39 | 5.18 |
| TokenHPE | × | 5.44 | **4.36** | 4.08 | 4.66 | 4.51 | **3.95** | **2.71** | **3.72** |
| 6DRepNet | × | **3.27** | 4.58 | **2.98** | **3.61** | **3.23** | 5.32 | 2.78 | 3.78 |
| WHENet | ✓ | 5.11 | 6.24 | 4.92 | 5.42 | 3.99 | 4.39 | 3.06 | 3.81 |
| 6DRepNet360 | ✓ | 3.58 | 5.28 | 3.46 | 4.11 | 3.28 | 6.06 | 3.08 | 4.14 |
| CLERF | ✓ | 4.22 | 6.18 | 4.67 | 5.02 | 3.57 | 4.49 | 3.13 | 3.73 |
| | | \multicolumn{4}{c}{SA AFLW2000} | | | SA BIWI | | |
| FSA-Net | × | 18.59 | 16.02 | 17.04 | 17.22 | 7.09 | 9.42 | 6.20 | 7.57 |
| HopeNet | × | 6.44 | 9.31 | 6.08 | 7.28 | 10.80 | 10.07 | 9.32 | 10.07 |
| TokenHPE | × | 6.29 | 7.29 | 6.56 | 6.70 | 6.84 | 7.12 | 5.19 | 6.39 |
| 6DRepNet | × | 8.41 | 8.80 | 7.68 | 8.30 | 6.45 | 7.09 | 6.79 | 6.78 |
| WHENet | ✓ | 13.11 | 12.88 | 15.06 | 13.68 | 9.51 | 10.99 | 9.52 | 10.00 |
| 6DRepNet360 | ✓ | 5.69 | 6.76 | 5.34 | 5.93 | 10.21 | 7.88 | 6.51 | 8.20 |
| CLERF | ✓ | **4.56** | **6.10** | **4.86** | **5.36** | **6.57** | **4.52** | **4.21** | **5.10** |
| | | \multicolumn{4}{c}{FA AFLW2000} | | | FA BIWI | | |
| WHENet | ✓ | 22.04 | 23.03 | 39.82 | 28.30 | 30.77 | 22.43 | 41.65 | 31.95 |
| 6DRepNet360 | ✓ | 14.00 | 16.93 | 21.96 | 17.63 | 25.97 | 17.90 | 34.04 | 25.97 |
| CLERF | ✓ | **4.84** | **5.79** | **4.31** | **4.98** | **7.68** | **7.89** | **6.93** | **7.50** |

Table 2: Comparison between supervised and contrastive learning. Contrastive learning yields a significant performance improvement over supervised training. Best results are highlighted in bold.

| Model | AFLW | | | | BIWI | | | |
|---|---|---|---|---|---|---|---|---|
| | Yaw | Pitch | Roll | Mean | Yaw | Pitch | Roll | Mean |
| CLERF-Supervised | 4.65 | 6.48 | 4.81 | 5.31 | 5.56 | 6.21 | **2.58** | 4.78 |
| CLERF | **4.22** | **6.18** | **4.67** | **5.02** | **3.57** | **4.49** | 3.13 | **3.73** |

performer among FR models, leading by more than 10° in Mean. Interestingly, CLERF shows a slight improvement of 0.03° on FA AFLW2000 compared to AFLW2000, reflecting its consistent optimization across the full pose range.

## 5.2 ABLATION STUDIES

**Contrastive learning.** To validate the benefit of contrastive learning, we compare it against a traditional supervised approach. The representation model is concatenated with the downstream MLP and trained using the geodesic loss on the same dataset. Results in Table 2 show a clear improvement with our contrastive learning framework. By grouping neighboring head poses while separating more distant ones, contrastive learning helps the MLP map representations to fine-grained head orientations more accurately.

**Batch size.** Batch size is an important hyperparameter in contrastive learning, as anchor–positives and anchor–negatives are sampled within each mini-batch. Due to the sparsity of head orientations, small batches may contain few valid positives for a given anchor. Using larger batches increases the likelihood of sampling additional positive examples while also providing a richer set of negatives.

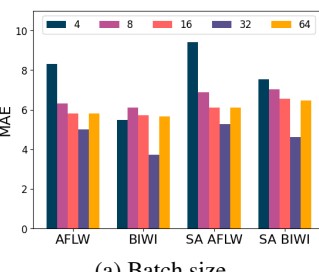 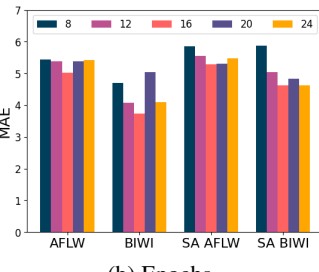 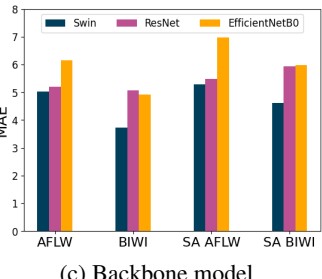

(a) Batch size      (b) Epochs      (c) Backbone model

Figure 2: Model performance under different choices of (a) batch size, (b) number of training epochs, and (c) image encoder backbone. The best performance is achieved with a batch size of 32, 16 training epochs, and the Swin Transformer backbone. (AFLW2000 is abbreviated as AFLW.)

Table 3: Ablation study on rotation and flip as data augmentation methods. Both individually improve performance on the original and slightly augmented (SA) test sets, while their combination achieves the best results. Best results are highlighted in bold. (Imp. % denotes the percentage improvement in Mean.)

| Augmentation | AFLW2000 | | SA AFLW2000 | | BIWI | | SA BIWI | |
|---|---|---|---|---|---|---|---|---|
| Method | Mean | Imp. % | Mean | Imp. % | Mean | Imp. % | Mean | Imp. % |
| CLERF | 6.88 | - | 9.93 | - | 5.48 | - | 8.99 | - |
| + Flip only | 5.98 | 13% | 7.78 | 22% | 4.96 | 9% | 7.08 | 21% |
| + Rotate only | 5.66 | 18% | 6.19 | 38% | 4.64 | 15% | 5.77 | 36% |
| + Rotate & Flip | **5.02** | **27%** | **5.36** | **46%** | **3.73** | **32%** | **5.10** | **43%** |

Figure 2(a) shows model performance across different batch sizes. We observe a consistent trend across all four datasets: performance improves with larger batch sizes up to 32, corresponding to 32 images from 300W-LP and 32 synthetic PanoHead images generated on-the-fly.

**Epochs.** We study the effect of training duration on the learned representations, with results shown in Figure 2(b). Model performance peaks at 16 epochs and declines thereafter, likely due to limited diversity in the synthetic images generated by PanoHead accumulating over repeated passes.

**Image encoder backbone.** While we use the Swin Transformer for its strengths, we compare it to two other backbones to evaluate our approach across different encoders, as shown in Figure 2(c). The Swin Transformer consistently outperforms the others, suggesting that larger backbones may be beneficial for this task.

**Rotation and flip augmentation.** Rotation and flip are key components of our approach that enable FR HPE. To assess their impact, Table 3 reports performance improvements for each. The combination of rotation and flip achieves the best results, while each contributes individually when applied alone. These geometric augmentations improve performance not only on AFLW2000 and BIWI but also on their SA variants.

## 5.3 CASE STUDIES AND VISUALIZATION

**Qualitative analysis.** To complement the quantitative results, we illustrate model performance with three test cases from the original, SA, and FA AFLW2000 test sets, comparing predictions to ground truth and baseline models (Figure 3). The first row shows an original image, where all models produce similar head pose predictions, consistent with the Mean values in Table 1. The second row depicts the SA version of the same image, where deviations are more noticeable for WHENet and TokenHPE. In the FA version (third row), CLERF accurately adapts to heavy rotations, while FR baseline models struggle. These visualizations highlight that CLERF consistently produces accurate FR predictions.

**TSNE visualization.** To further analyze the learned representations, we visualize embedding vectors with 3D t-distributed stochastic neighbor embedding (t-SNE) for a sequence of images showing a

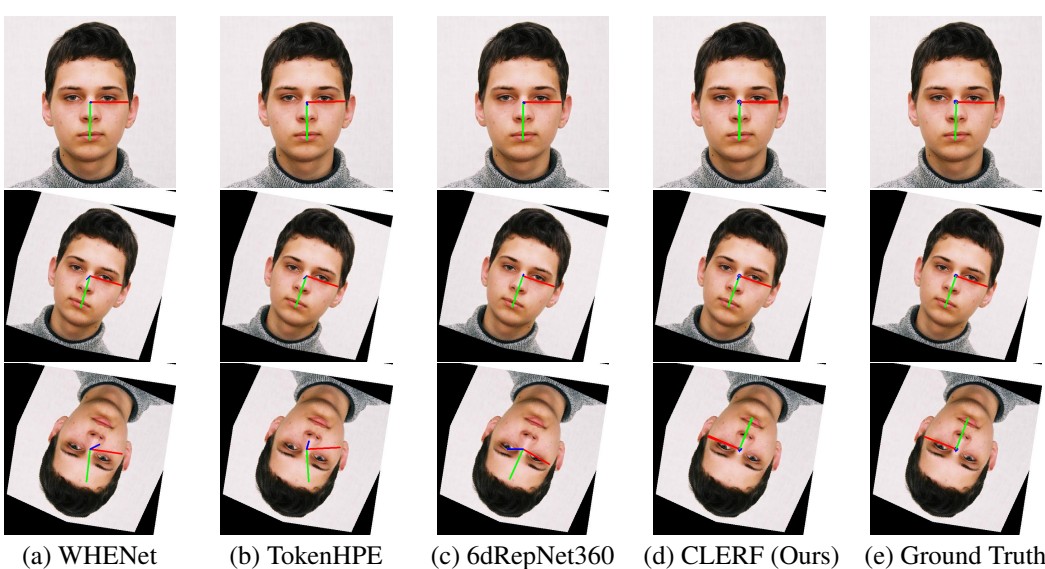

| (a) WHENet | (b) TokenHPE | (c) 6dRepNet360 | (d) CLERF (Ours) | (e) Ground Truth |

Figure 3: Head pose predictions of three baseline models (a–c) and CLERF (d), compared with ground truth (e), on the original image (first row), its slightly augmented (SA, second row), and fully augmented (FA, third row) versions. Head pose is visualized using three colored axes (red, blue, and green). While predictions are similar on the original and SA images, CLERF more accurately estimates head pose under the full-range (FA) transformation.

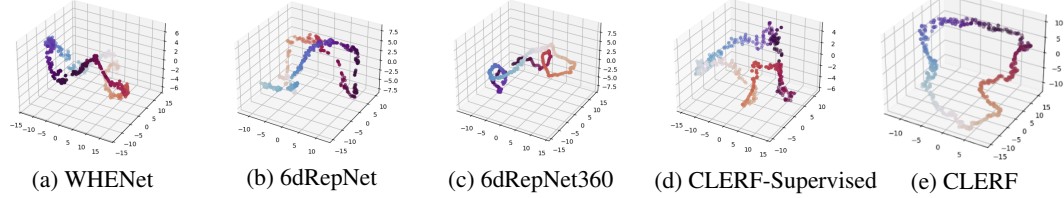

| (a) WHENet | (b) 6dRepNet | (c) 6dRepNet360 | (d) CLERF-Supervised | (e) CLERF |

Figure 4: 3D t-SNE visualization of embedding vectors from a video sequence across different models (a–e). The sequence shows a person completing a full 360° head rotation. Color similarity indicates temporal proximity between frames.

person turning in a full circle (Figure 4). Colors follow a cyclic map, where neighboring frames have similar colors; an optimal pattern forms a clear circle with colors changing continuously. CLERF exhibits strong separation between distant angles while keeping nearby angles close, as indicated by the smooth color transitions. In contrast, the supervised model without contrastive learning maintains local continuity but fails to separate opposite angles effectively. Similar patterns are observed for other baseline models.

## 6   CONCLUSION

The sparsity of head poses has previously limited the use of contrastive learning in head pose estimation. We proposed CLERF, a novel approach that leverages contrastive learning to separate representations of distant angles, improving head pose estimation performance. By generating anchor–positive pairs using a 3D-aware GAN combined with geometric transformations, our model achieves performance on par with state-of-the-art methods. While many prior works focus on frontal poses and suffer under slight rotations or flips, CLERF outperforms all baselines in such scenarios. Moreover, we demonstrate its full-range capability on heavily transformed images, achieving superior results compared to existing full-range models.

ACKNOWLEDGMENT

Xuyang Wu's contribution was made while he was an intern at DOCOMO Innovations.

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

## A    SPECIAL CASES OF GEOMETRIC FLIP TRANSFORMATIONS

**Corollary A.0.1.** *Special cases of the flip transformation in Equation  2 include:*

1. *Horizontal flip ($\theta = \pi/2$):*

$$R_{\text{hflip}} = \begin{bmatrix} -1 & 0 & 0 \\ 0 & 1 & 0 \\ 0 & 0 & 1 \end{bmatrix} \times R \times \begin{bmatrix} -1 & 0 & 0 \\ 0 & 1 & 0 \\ 0 & 0 & 1 \end{bmatrix} \tag{4}$$

2. *Vertical flip ($\theta = 0$):*

$$R_{\text{vflip}} = \begin{bmatrix} 1 & 0 & 0 \\ 0 & -1 & 0 \\ 0 & 0 & 1 \end{bmatrix} \times R \times \begin{bmatrix} -1 & 0 & 0 \\ 0 & 1 & 0 \\ 0 & 0 & 1 \end{bmatrix} \tag{5}$$

3. *Symmetric flip about the line $\ell_{\pi/4}$:*

$$R_{\text{diag}} = \begin{bmatrix} 0 & 1 & 0 \\ 1 & 0 & 0 \\ 0 & 0 & 1 \end{bmatrix} \times R \times \begin{bmatrix} -1 & 0 & 0 \\ 0 & 1 & 0 \\ 0 & 0 & 1 \end{bmatrix} \tag{6}$$

4. *Rotation by $\pi/4$ counter-clockwise:*

$$R_{\text{rot}}(\pi/4) = \begin{bmatrix} \cos(\pi/4) & -\sin(\pi/4) & 0 \\ \sin(\pi/4) & \cos(\pi/4) & 0 \\ 0 & 0 & 1 \end{bmatrix} \times R \tag{7}$$

## B    HEAD POSE DISTRIBUTION AFTER GEOMETRIC AUGMENTATION

Figure 5 shows the distribution of 300W-LP head poses after applying our rotation and flip transformations (red), compared to the original distribution (blue) and a set of random rotation matrices (green). Since 300W-LP lacks poses with large yaw angles (i.e., heads facing away from the camera), the transformed distribution does not fully cover the sphere, forming a semi-sphere rather than a complete sphere, as illustrated in Figure 5(c).

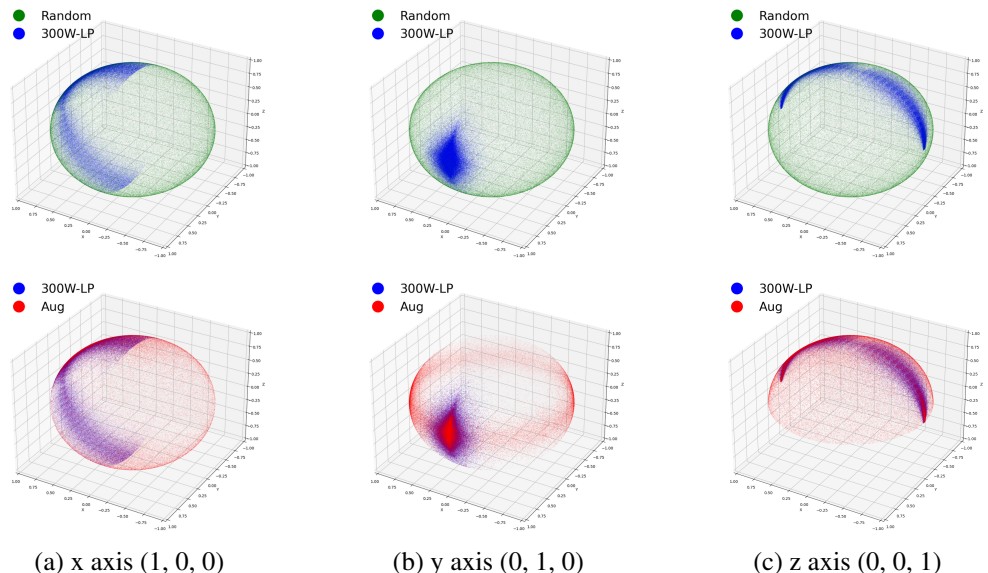

(a) x axis (1, 0, 0)           (b) y axis (0, 1, 0)           (c) z axis (0, 0, 1)

Figure 5: Visualization of the 300W-LP dataset after randomized rotation and flip augmentations. Each point on the sphere $\in \mathbb{R}^3$ represents a rotation matrix projected onto the three coordinate axes: (a) x, (b) y, and (c) z. The top row shows random rotation matrices (green) alongside the original 300W-LP poses (blue), while the bottom row shows the augmented 300W-LP poses (red). The geometric transformations expand the original dataset to provide wider head pose coverage.

## C  PROOF

The geodesic distance $d$ between two rotation matrices $A$ and $B$ is defined as (Hempel et al., 2024):

$$d(A, B) = \cos^{-1}\left(\frac{\text{tr}(AB^T) - 1}{2}\right), \tag{8}$$

where $\text{tr}$ and $T$ denote the trace and transpose operators, respectively.

*Proof.* Suppose $A$ and $B$ are rotation matrices $\in SO(3)$ and $\theta \leq 90°$. According to Equation 2, flipping across $\ell_\theta$ on $A$ and $B$ are $A_{flip}(\theta) = Flip\_\theta \times A \times Flip\_X$ and $B_{flip}(\theta) = Flip\_\theta \times B \times Flip\_X$. From Equation 8, $tr(A \times B^T)$ is the unique factor that can alter the geodesic distance. To show that the geodesic distance between $A_{flip}(\theta)$ and $B_{flip}(\theta)$ is equal to the geodesic distance between $A$ and $B$, it suffices to show the equality $tr(A \times B^T) = tr(A_{flip}(\theta) \times B_{flip}(\theta)^T)$ holds. Let's consider $A_{flip}(\theta) \times B_{flip}(\theta)^T$ first:

$$\begin{aligned}
A_{flip}&(\theta) \times B_{flip}(\theta)^T \\
&= (Flip\_\theta \times A \times Flip\_X) \times (Flip\_\theta \times B \times Flip\_X)^T \\
&= Flip\_\theta \times A \times Flip\_X \times Flip\_X^T \times B^T \times (Flip\_\theta)^T \\
&= Flip\_\theta \times A \times (Flip\_X \times Flip\_X^T) \times B^T \times (Flip\_\theta)^T \\
&= Flip\_\theta \times A \times B^T \times (Flip\_\theta)^T
\end{aligned} \tag{9}$$

Next, the commutativity of the trace of a matrix guarantees the following:

$$\begin{aligned}
tr(A_{flip}(\theta) \times B_{flip}(\theta)^T) &= tr(Flip\_\theta \times A \times B^T \times (Flip\_\theta)^T) \\
&= tr(A \times B^T \times (Flip\_\theta \times Flip\_\theta^T)) \\
&= tr(A \times B^T).
\end{aligned} \tag{10}$$

$\square$

Therefore, flipping preserves the geodesic distance: $d(A, B) = d(A_{flip}(\theta), B_{flip}(\theta))$. Similarly, we can apply Equation 1 to prove that the equality $tr(A, B) = tr(A_{rotated}(\phi), B_{rotated}(\phi))$ holds for the rotations associated with rotating an image by an angle $\phi$ based on $A$ and $B$, so rotation preserves the geodesic distance as well.

## D  EXAMPLES FOR ANCHOR-POSITIVE SYNTHETIC IMAGE GENERATION

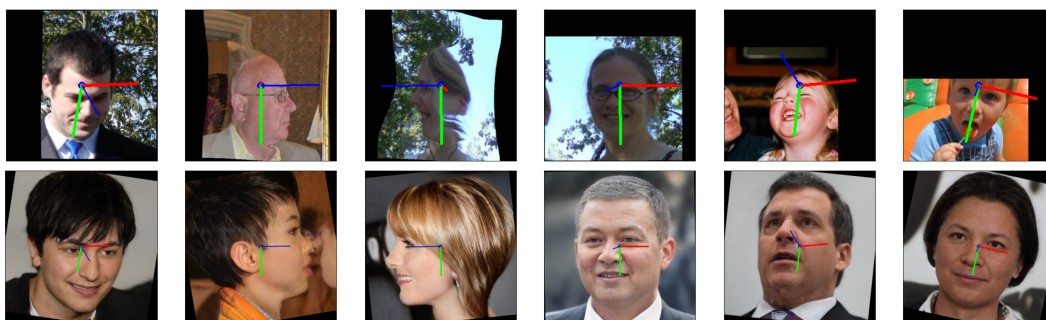

Figure 6: Illustration of synthetic positive generation for contrastive learning. The first row shows rotation matrices corresponding to images from the 300W-LP dataset. The second row shows synthetic images generated to match the same head poses. In the contrastive learning setting, images in the first row serve as anchors, those in the second row as anchor-positives, while anchor-negatives can be readily sampled due to the sparsity of head poses in the pose space.

# E IMPLEMENTATION DETAILS

**Representation model.** We adopt the Swin Transformer Base model (Liu et al., 2022) pre-trained on ImageNet as our image encoder $E$, producing embeddings of size 1024. The model is trained for 30 epochs on 20,000 images from 300W-LP and 20,000 synthetic images generated by PanoHead, using the Adam optimizer with a learning rate of $10^{-5}$ on a single NVIDIA Tesla V100 32GB GPU. Triplet sampling and loss computation are implemented via the PyTorch Metric Learning library (Musgrave et al., 2020). For hyperparameters, we set $T_{GD} = 0.8$ and $v = 0.1$ for triplet filtering, and $m = 0.4$ and $\gamma = 80$ under Circle loss.

**MLP.** Our downstream MLP consists of four fully connected layers with 256 units each and a skip connection from the input to the final layer. It is trained for 40 epochs using the mean geodesic distance $d$ as the loss, with an exclusive subset of the training data used as a validation set for early stopping.

**Data augmentation.** For geometric augmentation, each image is randomly rotated clockwise by 0–90° with probability 0.5 and flipped along an axis between 0–90° counter-clockwise from the positive $x$-axis with probability 0.3. Pixel-wise augmentations include translation, resizing, down-sampling, hue adjustment, sharpness, grayscale conversion, contrast-limited adaptive histogram equalization, brightness adjustment, RGB shift, channel shuffle, gamma correction, color jitter, Gaussian noise, and Gaussian blur. For pixel removal, we apply center cropping and coarse dropout. These augmentations alter both spatial and pixel-level properties, improving robustness during training.

**Baseline models and evaluation metric.** We compare our approach against several existing HPE models. FSA-Net (Yang et al., 2019) uses feature aggregation with soft stagewise regression, HopeNet (Ruiz et al., 2018) applies multiple losses on a convolutional neural network, and TokenHPE (Zhang et al., 2023) leverages a transformer to model relationships between facial parts. 6DRepNet (Hempel et al., 2024) employs a geodesic loss, with 6DRepNet360 extending it to full-range head poses. WHENet (Zhou & Gregson, 2020), another full-range model, wraps the loss function to stabilize learning for large yaw angles. Following prior work, we evaluate all models using mean absolute error (MAE) for yaw, pitch, and roll, and report the average of the three (Mean) as the primary metric.

