# OpenReview forum: "CLERF: Contrastive LEaRning for Full-Range Head Pose Estimation"
_ICLR.cc/2026/Workshop/GRaM — ICLR 2026 Workshop GRaM Poster_

### Official Review · Reviewer_QqZc · 2026-02-11
**Well-executed applied work, but geometry contribution is thin for GRaM**

**Rating:** 6
**Confidence:** 4

**Review:**

Review:Summary. CLERF proposes a contrastive learning framework for full-range head pose estimation that uses a 3D-aware GAN (PanoHead) to generate synthetic anchor-positive pairs, combined with geometric augmentations (rotations and flips with corresponding SO(3) transformations) to cover the full rotation space. The method trains a Swin Transformer encoder with Circle loss, then freezes it and trains a downstream MLP for pose regression. CLERF matches state-of-the-art on standard frontal benchmarks (AFLW2000, BIWI) and substantially outperforms existing full-range models on augmented test sets.Strengths.The paper identifies a genuine and well-articulated problem: contrastive learning is difficult for HPE because positive pairs with similar 3D orientations are vanishingly rare in real datasets. The probability calculation (1/18³ ≈ 2×10⁻⁴ at 20° tolerance) concretely motivates the approach. Using a 3D-aware GAN to synthesize matched anchor-positives is a clean solution to this problem.The geometric transformation framework is carefully derived. The general flip formula (Equation 2) parameterized by line angle θ is elegant, the special cases in Appendix A are correctly worked out, and Theorem 3.1 proving geodesic distance preservation under these transformations is correct and relevant — it formally guarantees that augmented triplets remain valid. The proof in Appendix C is straightforward but sound.The experimental results are convincing where it matters most. On fully-augmented test sets, CLERF achieves 4.98° and 7.50° mean MAE versus 17.63° and 25.97° for the next-best full-range model (6DRepNet360) — improvements exceeding 10° that are unambiguously significant. The ablation studies are well-structured: Table 2 cleanly isolates the contrastive learning benefit over supervised training, Table 3 decomposes the contributions of rotation and flip augmentations, and the batch size / epoch / backbone sweeps (Figure 2) are informative. The t-SNE visualization (Figure 4) provides compelling qualitative evidence that CLERF's embedding space respects rotational structure more faithfully than baselines.The paper is clearly written, well-organized, and honest about its limitations (e.g., acknowledging that CLERF trades slight frontal-range accuracy for full-range coverage).Weaknesses.The central concern for this venue is whether the paper makes a sufficient contribution to geometry and representation learning, as opposed to being a well-executed application of standard geometric principles to a specific CV task. The geometric content — SO(3) transformations, geodesic distance preservation, isometry proofs — is correct but textbook-level. Theorem 3.1 follows immediately from the invariance of trace under cyclic permutation and the orthogonality of the transformation matrices; labeling this a "theorem" somewhat overstates its novelty. The GRaM workshop description emphasizes advancing understanding of geometric and relational methods, and while this paper uses geometry, it does not contribute new geometric insights or methods.The evaluation protocol has a significant circularity issue. The SA and FA test sets are created by applying the exact same geometric transformations (rotation and flip with corresponding SO(3) updates) that CLERF is trained with. Baseline models were never designed or trained to handle these transformations, making the comparison inherently unfair. The massive margins on FA datasets (>10° improvement) primarily demonstrate that baselines lack equivariance to these specific transformations — which is expected and not particularly informative. A fairer evaluation would test on naturally occurring extreme poses (e.g., from sports, acrobatics, or surveillance footage), which the paper mentions as motivation but never actually evaluates on.The synthetic-to-real domain gap is inadequately addressed. PanoHead-generated images (Figure 6, Appendix D) have visible artifacts and differ substantially from real photographs in texture, lighting, and background. The paper uses these synthetic images as anchor-positives during contrastive learning but does not analyze or control for the domain gap. If the encoder learns to distinguish real from synthetic images in embedding space, the contrastive objective could be partially corrupted — pushing apart real-synthetic pairs that should be pulled together. No analysis of this failure mode is provided (e.g., measuring embedding distances between real and matched synthetic images).The paper does not report variance or confidence intervals for any result. Table 1 presents single numbers with no indication of statistical reliability. Given that contrastive learning is known to be sensitive to batch composition and initialization, this is a notable omission. How stable are the results across random seeds?The claim that CLERF achieves "on par" performance with state-of-the-art on standard benchmarks requires qualification. On AFLW2000, CLERF trails 6DRepNet by 1.4° in Mean (5.02 vs. 3.61) — this is a 39% relative increase in error, which is substantial. On BIWI, the 0.01° gap with TokenHPE (3.73 vs. 3.72) is indeed negligible. The paper somewhat glosses over the AFLW2000 gap by framing it as "trailing by only 1.4°," but in a field where state-of-the-art improvements are often measured in fractions of a degree, this is a meaningful regression.The downstream architecture (frozen encoder + 4-layer MLP) means that fine-grained pose discrimination depends entirely on the contrastive representations being sufficiently expressive. The paper does not analyze whether the bottleneck is in the representation or the MLP. An experiment varying MLP capacity or unfreezing the encoder for joint fine-tuning would help characterize this.The paper evaluates only on Euler-angle MAE, which has known issues: it weights errors near gimbal lock differently than errors in well-conditioned regions, and it treats yaw/pitch/roll independently despite their coupled nature. Given the emphasis on SO(3) geometry, reporting geodesic distance error directly would be more principled and internally consistent with the training loss.

**Pmlr Suitability:**

Yes

---

### Official Review · Reviewer_cFh1 · 2026-02-20
**Promising Contrastive HPE with limitations**

**Rating:** 6
**Confidence:** 4

**Review:**

The authors propose a contrastive learning method for full-range head pose estimation. The framework leverages a 3D-aware GAN ( PanoHead), to generate controlled poses with  anchor-positive-negative triplets to perform contrastive learning on rotation-augmented data.

Strengths:
The general problem of full range head pose estimation is clearly identified and well motivated on why standard HPE models struggle under rotations and flips and why full-range behavior matters.
The paper is well written overall and experiments make sense to support the approach
Using synthetic positives addresses the main bottleneck of having enough true positives from real pose datasets
I like the t-SNE visualization supporting the quality of predicted embeddings regarding pose continuous changes

Major comments/remarks/questions:
- The approach is ultimately bound to the efficiency of the 3D-aware GAN. In particular, PanoHead is known to exhibit Janus artifacts (see CylinderPlane: Nested Cylinder Representation for 3D-aware Image Generation for more), have you observed such issues ? Can it be resolved with other models ?
- The paper claims “ To our knowledge, contrastive learning has not been applied to HPE”, but there is “Head pose estimation with particle swarm optimization-based contrastive learning and multimodal entangled GCN” which already uses contrastive learning for this task
- The FA/SA protocols are built via image-plane rotations and flips with corresponding pose-label updates, ok, but it is different from true 3D full-range viewpoint changes with, for example, yaw beyond +-90° that reveals the back of the head and severe self-occlusion.

Minor comment:
- Organisation of the paper: Section 4 is “Experiments” and Section 5 is “Experimental results”. I suggest renaming or unifying under one section.

**Pmlr Suitability:**

Yes

---

### Official Review · Reviewer_fEEd · 2026-02-23
**Strong Full_Range Contribution with Strong Experiments, but Limited Real-World Validation**

**Rating:** 6
**Confidence:** 4

**Review:**

**Strengths:**
- The problem is well motivated. The sparsity of head pose samples in full 3D rotation space is a clearly articulated bottleneck for contrastive learning and the use of 3D-aware GANs to generate anchor-positive pairs is a strong practical solution.
- Strong and comprehensive experiments with multiple test conditions of various augmentations across two datasets, demonstrating the distinct gains of CLERF achieved over baselines in the range of scenarios.

**Weakness:**
- There is a consistent small, but non-trivial performance drop on standard non-augmented AFLW2000 and BIWI test sets compared to the best non-FR baselines.
 - Synthetic images from PanoHead are used only for representation training without any independent evaluation of their pose accuracy or visual realism, leaving open the question of how label noise in the synthetic anchor-positive pairs affect the quality of the learned representations.
- No evaluation on in-the-wild full-range data makes it unclear where the performance gain reflects an inherent learned capability or an artifact of learning the transformation distinctly with respect to the training distribution.

**Questions:**
- What accounts for the consistent performance drop on the non-augmented test sets? Is there an inherent trade-off in optimizing for performance on non-augmented vs augmented poses?
- How does CLERF perform on truly in-the-wild full-range data beyond the augmentation AFLW200 and BIWI splits, such as sports or acrobatic footage mentioned in the introduction?

**Pmlr Suitability:**

Yes

---

### Meta-Review · Area_Chair_RZzg · 2026-02-27

**Decision:**

Accept

**Metareview:**

This work presents a contrastive learning method for full-range head pose estimation. I believe it is a fit for GRaM proeceedings and would like to highlight the concerns of reviewers.

Reviewers agree on acceptance with a concerns like misrepresenting previous work on contrastive methods, lack of evaluation on Panohead, misattributing the novelty of Theorem 3.1 being an important one.

I highly recommend the authors to correct their manuscripts by answering the the weaknesses and concerns listed in the reviews below for the camera ready version.

**Relevance To Proceedings:**

Yes — suitable for PMLR (long paper)

**Relevance To Workshop:**

Yes — suitable for GRaM

---

### Decision · Program_Chairs · 2026-03-02

Accept (Poster)